# The First Evidence of the Insecticidal Potential of Plant Powders from Invasive Alien Plants against Rice Weevil under Laboratory Conditions

**Tanja Bohinc [1,*], Aleksander Horvat [2], Miha Ocvirk [3], Iztok Jože Košir [3] , Ksenija Rutnik [3] and Stanislav Trdan [1]**

1  Department of Agronomy, Biotechnical Faculty, University of Ljubljana, Jamnikarjeva 101,
   SI-1000 Ljubljana, Slovenia; stanislav.trdan@bf.uni-lj.si
2  Scientific Research Centre of the Slovenian Academy of Sciences and Arts,
   Ivan Rakovec Paleontological Institute, Novi trg 2, SI-1000 Ljubljana, Slovenia; ahorvat@zrc-sazu.si
3  Slovenian Institute of Hop Research and Brewing, Chemical Analysis and Brewing, Cesta Žalskega tabora 2,
   SI-3310 Žalec, Slovenia; miha.ocvirk@ihps.si (M.O.); iztok.kosir@ihps.si (I.J.K.); ksenija.rutnik@ihps.si (K.R.)
*  Correspondence: tanja.bohinc@bf.uni-lj.si

**Abstract:** In a laboratory experiment, we studied the insecticidal effects of invasive alien plants on the rice weevil. The research was carried out in two parts. In the first part, we studied the insecticidal properties of seven different plant species, namely, Bohemian knotweed (*Fallopia × bohemica*), Japanese knotweed (*Fallopia japonica*), false indigo-bush (*Amorpha fruticosa*), tree of heaven (*Ailanthus altissima*), staghorn sumac (*Rhus typhina*), Canada goldenrod (*Solidago canadensis*), and giant goldenrod (*Solidago gigantea*). Mixtures of powders and wheat were prepared in two different concentrations, namely, 2.5 w% and 1.25 w%. The experiment was performed at temperatures 20 °C and 25 °C and at two humidity levels, 55% R.h. and 75% R.h. Very low mortality (below 8%) was found when using combinations with the higher relative humidity. No significant differences were observed between the effects of these concentrations. In the second part of the experiment, Norway spruce wood ash and diatomaceous earth (product SilicoSec®) were added to the powder obtained by milling leaves of four different invasive plant species (Canada goldenrod, staghorn sumac, tree of heaven, false indigo). In the independent application, wheat was added to the powder at a concentration 2.5 w%. In the treatments that involved mixtures of powder and wood ash/diatomaceous earth, we applied 1.25 w% plant powder and 1.25 w% wood ash or 1.25 w% plant powder and 450 ppm of a SilicoSec®preparation. The positive control was carried out as two separate treatments with 2.5 w% wood ash of Norway spruce and 900 ppm of the SilicoSec® product, while untreated wheat represented the negative control. The experiment was performed at two temperatures (20 °C and 25 °C) and two R.h. values (55 and 75% R.h.). The mortality of beetles was recorded on the 7th, 14th, and 21st day after the start of the experiment. Higher mortality rates of rice weevil adults were found at the higher relative humidity, and an important factor of mortality was also the day of exposure, as a higher mortality was found when the exposure of individuals to the tested substances was for a longer time period. After 21 days at 25 °C and 55% R.h., the combinations in which the lower concentration of Norway spruce wood ash was added to the powder of invasive alien plants achieved more than 90% mortality of beetles. By adding the plant powder of invasive alien plants to wood ash, we achieved a greater insecticidal efficacy of invasive plants and lower concentrations of wood ash. Nevertheless, the results of our research do not indicate any great usefulness of the plant powder of invasive plants in suppressing the rice weevil. Additional studies should primarily focus on the insecticidal efficacy of powder from the genus *Solidago*, which in our study, displayed the greatest insecticidal potential among the tested invasive plants.

**Keywords:** rice weevil; invasive alien plant species; tree of heaven; staghorn sumac; false indigo; Japanese knotweed; Bohemian knotweed; giant goldenrod; Canada goldenrod; wood ash; diatomaceous earth; synergistic effect; Applause

---

## 1. Introduction

The most researched alternative method of suppressing storage insect pests in the last 20 years has been the use of inert dusts, which include, among other substances, diatomaceous earth and wood ash. The use of diatomaceous earth has numerous advantageous characteristics, as well as some undesired ones [1]. To reduce the negative effects of diatomaceous earth, such as reducing the density of wheat and affecting its pourability [2], many studies have focused on the synergistic effects of plant insecticides and diatomaceous earth [1]. Research on the effects of wood ash in the suppression of storage insect pests was described by [3,4]. The plant from which the most insecticides have been obtained to date is neem (*Azadirachta indica* A. Juss) [5]. Bioactive substances and plant insecticides can work in many ways: they may act as repellents, affect oviposition or feeding, cause disruptions in development, or produce acute mortality of insect pests [6]. The most frequently mentioned among the plant species suitable for producing plant insecticides are *Chrysanthemum cinerarifolium*, *Rosmarinus officinalis*, *Nicotiana* sp., etc. [6]. The search for alternative methods for the suppression of insect pests is very important, as total food production may plummet by 70% due to the decreasing number of available synthetic insecticides and the fact that at the moment some 67,000 species of organisms endanger food production [7]. Beetles from the genus *Sitophilus*, to which the rice weevil (*Sitophilus oryzae* [L.]) also belongs, are a group of pests whose feeding causes damage to stored crops (grains) all over the world [8].

The plants used in our study are found in Slovenia and Europe and are on the list of invasive alien plants. These plants have a negative impact on the environment in which they appear and cause damage to the global economy. These plants are also difficult to remove from the environment since their suppression does not always prove efficient [9]. Studies on the usefulness and efficacy of invasive alien plants as plant insecticides are scarce [10]. Satisfactory molluscicidal properties of milled leaves of Canada goldenrod were reported [11]. Therefore, our aim is to find additional uses of invasive alien plants, which is important in Slovenia.

## 2. Materials and Methods

### 2.1. Collection and Preparation of Plant Material

In the area of the Ljubljana municipality (46°03′ N, 14°31′ E, 299 above sea level), we collected leaves and flowers from seven different invasive alien plant species. Leaves were collected from Canada goldenrod (*Solidago canadensis* L.; collected on 28 August 2018), giant goldenrod (*Solidago gigantea* L.; collected on 24 August 2018), Japanese knotweed (*Fallopia japonica*/Houtt./Ronse Decr.; collected on 13 August 2018), Bohemian knotweed (*Fallopia × bohemica* (Chrtek & Chrtkova) Bailey; collected on 6 July 2018), false indigo (*Amorpha fruticosa* L.; collected on 23 July 2018), tree of heaven (*Ailanthus altissima*/Mill./Swingle; collected on 6 September 2018), and staghorn sumac *(Rhus typhina* L.; collected on 4 July 2018). Flowers were collected only from Canada goldenrod and giant goldenrod. All tested invasive plant species were collected in all given dates. Due to the fact that several research works were done in the same time interval, this research contained material from several days.

The collected plant material was dried at air temperature in a shadowed place in a warehouse in the Department of Agronomy. After three weeks, the dried material was milled with a mill (type: 880803\, producer: Brabender GmbH & Co. KG, Duisburg, Germany). Plant powders were stored in plastic boxes and stored in a freezer (type: U3286S; producer: Sanyo, Krimpen aan den IJssel, Netherlands) at −80 °C.

Particle size of wood ash and plant powders was ranging from 20–200 μm.

### 2.2. Chemical Analysis of Plant Material

Active ingredients that were detected in our survey were chosen according to detailed study of literature and evidence of possible insecticidal efficacy.

#### 2.2.1. Preparation of Sample Extracts

For the determination of phenolic substances, ultrasonic extraction was performed on finely powdered plant tissue (1 g) with 25 mL of methanol (manufacturer: Sigma-Aldrich, St. Louis, MO, USA) at 25 °C for 60 min. The sample extracts were filtered through a 0.45 μm PTFE 25 mm filter (Restek).

#### 2.2.2. Preparation of Standard Solutions

Stock solutions of standard compounds (quercetin, rutin, caffeic acid, naringin, ferulic acid, hydroxy-coumarin, catechyn hydrate, and p-coumaric acid) (all purchased at Sigma-Aldrich) were diluted separately in a mixture of water and methanol (1:1, *V/V*) at a concentration of 1.0 mg/mL. Working standard solutions at a concentration of 0.01 mg/mL were made by diluting each stock solution with the same mixture of water and methanol. Additionally, a mixture of standards was created by diluting each stock standard solution with the same solvent mixture at a final concentration of each standard of 0.01 mg/mL.

#### 2.2.3. HPLC Conditions

HPLC (high–performance liquid chromatography) analysis was performed using an Agilent 1100 Series HPLC system (Agilent Technologies, Santa Clara, CA, USA). A $C_{18}$ reversed-phase packing column (YMC Triart $C_{18}$, 150 mm × 4.6 mm, 5 μm; Agilent Technologies, Santa Clara, CA, USA) was used for separation at a temperature of 25 °C. Gradient elution chromatographic systems were used. The mobile phase for gradient elution consisted of solvent A (acetonitrile) and solvent B (5 mM $CH_3COOH$ in water), which were applied according to the following program: start with 25% A and 75% B for 10 min, followed by a gradient from 25% A to 100% A over 20 min, then to 250% A over 10 min. In the next 5 min, the initial conditions were set up. The injection volume of standards and plant samples was 20 μL. The flow rate was 1.0 mL/min. Detection was performed by a diode array detector at wavelengths of 280 nm for naringin, catechin, and hydroxyl coumarin; 320 nm for caffeic acid, ferulic acid, and coumaric acid; 370 nm for quercetin and rutin; and 254 nm for quercitrin.

Identification of particular compounds was achieved by comparing the retention times of the standards and unknown peaks in the samples. To avoid misinterpretation of results, the method of standard addition was applied. Quantification was conducted using external standards.

#### 2.2.4. GC Analysis

Determination of the total essential oil content was carried out according to the Analytica–EBC 7.10 method. Briefly, 100 g of dry and ground plant tissues was mixed with 1000 mL of deionized water and steam distilled for 3 h using a Clevenger distillation unit. Identification and quantification of the components of essential oils were carried out according to the Analytica–EBC 7.12 method. First, 0.1 mL of collected oil was diluted with 2 mL of hexane and separated by GC analysis. An Agilent 6890 series GC system equipped with a flame ionization detector and a HP-1 capillary column (30 m × 0.25 mm, 25 μm, manufacturer: Agilent Technologies, city: Santa Clara, CA, USA) with 5.0 helium as the carrier gas with a flow rate of 0.5 mL min$^{-1}$ was used. One microliter of solution was injected in the injector at a temperature of 200 °C. The temperature programme was 1 min at 60 °C, 2.5 °C min$^{-1}$ to 190 °C, 70 °C min$^{-1}$ to 240 °C, and 11 min at 240 °C. Detection was carried out on a flame ionization detector set at 260 °C. All solvents were of analytical grade or higher purity and were purchased from Sigma-Aldrich, Germany. The identified and quantified components were α-pinene, camphene, sabinene, β-pinene, β-myrcene, α-phellandrene, p-ocymene, limonene, trans ocymene, linalool, nonanal, borneol, α-terpineol, dodecane, bornyl

acetate, α-cubebene, α-copaene, β-bourbonene, β-elemene, cyprene, β-caryophyllene, α-caryophyllene, γ-gurjunene, β-copabene, σ-cadinene, bicyclogermacrene, germacrene-D, β-selinene, β-ionene, α-selinene, α-murolene, γ-cadinene, β-bisabolene, γ-murolene, isoledene, β–copaene, γ-gurjunene, α-bergamotene, β-seline, cadina-1(10,4-diene, β–sesquiphellandrene, germacren B, spathulenol, aromaderdrene, β–turmerone, cyperone, farnesyl acetone, 1,2-hexadecen-1-ol,3,7,11,15-tetramethyl, and 3,7,11,15-tetramethyl-2-hexadecen-1-ol. For some components, identification was performed with the use of standards by standard addition and through comparison of retention times. For those that were not available, identification was performed by the use of GC/MS and the NIST library of mass spectra with a probability of more than 80%. Standard compounds used for identification were purchased from Sigma-Aldrich, Germany.

### 2.3. Test Insects and Commodity

Rice weevil adults were maintained at room temperature (22 ± 2 °C) and relative humidity (R.h.) (55 ± 5%) in continuous darkness. Beetles were maintained for three days at the Laboratory of Entomology, Chair for Phytomedicine, Agricultural Engineering, Field Crop Production, Pasture and Grassland Management, Biotechnical Faculty, Ljubljana. Thirty rice weevils were purchased from the Pesticide and Environment Research Institute (Belgrade, Serbia). Rice weevil adults were 2–4 weeks old. Bioassay 1 and bioassay 2 were performed on untreated wheat, variety 'Olimpija'.

### 2.4. Geochemical Analysis of Wood Ash

Geochemical analysis of tested wood ash was carried out according to the methodology previously described by [4].

### 2.5. Admixture of Plant Powders for Single Use (Bioassay 1)

The first part of our research was based on ten different plant powders obtained from seven different invasive plant species. We used plant powders that were obtained from leaves of false indigo, Japanese knotweed, Canada goldenrod, giant goldenrod, tree of heaven, staghorn sumac, and Bohemian knotweed. We also prepared plant powders from flowers of giant goldenrod and Canada goldenrod and fruits from staghorn sumac. As a positive control, we used wood ash obtained from Norway spruce (*Picea abies*; vicinity of Jesenice, 46°21′56.64″ N 14°18′31.37″ E, 516 m height above sea level).

Erlenmeyer flasks (1000 mL) were filled with 500 g of winter wheat. The efficacy of the plant powders and wood ash was tested at two different rates, 2.5% and 1.25 w% of plant powder per grain weight. The preparation of each individual treatment was conducted according to the methodology described by [4]. Thirty individuals were placed into 60-mL flasks. Flasks were covered with mesh to prevent rice weevil adults from escaping. Untreated wheat served as the control treatment. The bioassay was performed at two different temperatures (20 °C and 25 °C) and two different relative humidities (R.h.; 55 and 75%). All bioassays were repeated three times. Mortality counts were performed after the 7th, 14th, and 21st days of exposure. Determining the number of offspring was not within the scope of bioassay 1.

### 2.6. Admixture of Plant Powder for Combined Use with Inert Dust (Bioassay 2)

Erlenmeyer flasks (1000 mL) were filled with 270 g of winter wheat. This study was based on 15 different treatments that included combined plant powders (from leaves) of four invasive plant species (tree of heaven, false indigo, Canada goldenrod, staghorn sumac) with two inert dusts, wood ash from Norway spruce and diatomaceous earth, product SilicoSec® (in figures marked as DE) (producer: Biofa, Münsingen, Germany; supplier: Metrob Ltd., Začret, Slovenia). The single use of the plant powders was also tested. All combinations with their concentrations are presented in Table 1. Particle size of diatomaceous earth was ranging from 2–18 μm. The bioassay was also performed at two different temperatures (20 °C and 25 °C) and two different R.h. values. Mortality counts were

performed after the 7th, 14th, and 21st days of exposure. We again used the same methodology described by [4,12]. Twenty rice weevil adults were placed into 60-mL flasks, and the untreated control served as one of the control treatments. After 56 days, we counted the number of offspring (progeny).

**Table 1.** Treatments used in our survey.

| Plant Powder | Dose |
|---|---|
| Tree of heaven | 2.5 w% |
| False indigo | 2.5 w% |
| Canada goldenrod | 2.5 w% |
| Staghorn sumac | 2.5 w% |
| Tree of heaven × wood ash | 1.25 w% × 1.25 w% |
| False indigo × wood ash | 1.25 w% × 1.25 w% |
| Canada goldenrod × wood ash | 1.25 w% × 1.25 w% |
| Staghorn sumac × wood ash | 1.25 w% × 1.25 w% |
| Tree of heaven × SilicoSec® | 1.25 w% × 450 ppm |
| False indigo × SilicoSec® | 1.25 w% × 450 ppm |
| Canada goldenrod × SilicoSec® | 1.25 w% × 450 ppm |
| Staghorn sumac × SilicoSec® | 1.25 w% × 450 ppm |
| Control wood ash | 2.5 w% |
| Control SilicoSec® | 900 ppm |
| Control–untreated grain | |

*2.7. Data Analysis*

The acquired mortality data were adjusted for mortality in the control using a previously described formula [13] when it exceeded 5%, and the data are expressed as percentages. Mortality counts were analyzed by using repeated measures MANOVA (multivariate analysis of variance) with exposure day as the repeated measures variable and treatment, temperature and R.h. as the main effects. Progeny production was analyzed by using one-way ANOVA (analysis of variance) to determine the effects of treatment, temperature, and R.h. The mean mortality counts (all bioassays) were separated by using the Tukey HSD (honestly significant difference) test at the 5% level [14].

## 3. Results

*3.1. Chemical Determination of Essential Oil (as mL/100 g Sample)*

The highest amount of essential oil was detected in the sample obtained from giant goldenrod leaves. In more detail, the level of beta-copabene was the highest in the sample from giant goldenrod leaves. Beta-copabene was only detected in samples from giant goldenrod, as well as in flowers, where the level reached 68%. Delta-canidine was only detected in samples from Canada goldenrod leaves. The levels of essential oil from staghorn sumac leaves and fruits, Bohemian knotweed, and tree of heaven were very low (0.01–0.04 mL/100 g sample), so they are not presented in Table 2 with all the other invasive alien plants. We did not detect essential oils from the sample from Japanese knotweed.

**Table 2.** Amount of essential oil in samples, presented as the rel. %.

| | Canada Goldenrod Leaves | Canada Goldenrod Flowers | Giant Goldenrod Leaves | Giant Goldenrod Flowers | False Indigo Leaves |
|---|---|---|---|---|---|
| amount of oil (mL/100 g) | 2.01 | 0.69 | 2.18 | 1.12 | 0.40 |
| alfa-pinene | 1.86 | 1.70 | 2.23 | 0.80 | 1.27 |
| camphene | 0.12 | * | 0.48 | * | * |
| sabinene | * | * | 0.12 | * | * |
| beta-pinene | 0.40 | 0.25 | 0.74 | 0.11 | 0.20 |
| beta-myrcene | 0.24 | 0.37 | 0.18 | 0.17 | 0.28 |
| alpha-phellandrene | * | * | 0.31 | * | * |
| p-cymene | * | * | 0.08 | 0.01 | * |
| limonene | 0.69 | 1.90 | 0.57 | 0.56 | 0.11 |
| trans ocimene | * | * | * | * | 1.20 |
| ocimene | * | * | * | * | 1.60 |
| borneol | 0.39 | 0.23 | 0.08 | 0.19 | * |
| bornyl acetate | 1.26 | 2.59 | 4.02 | 1.51 | * |
| alpha-cubebene | * | * | * | * | 0.69 |
| alpha-copaene | 0.12 | 0.26 | 0.09 | 0.21 | 1.39 |
| beta-bourbonene | 0.34 | 0.19 | 0.23 | * | * |
| beta-elemene | 1.42 | 1.40 | 0.13 | 1.69 | 0.57 |
| cyprene | 0.47 | 0.24 | 0.62 | 0.63 | 0.39 |
| beta-caryophyllene | 1.54 | 2.24 | 9.92 | 1.95 | 5.11 |
| alpha-caryophyllene | 0.58 | 1.02 | 0.43 | 1.04 | 1.33 |
| gama-gurjurene | * | * | 2.42 | * | * |
| beta-copabene | * | * | 56.44 | 68.31 | * |
| delta-cadinene | 54.31 | * | * | * | * |
| bicyclogermacrene | 0.81 | 1.27 | 2.57 | 1.40 | * |
| germacrene D | * | * | * | * | * |
| beta-selinene | 2.40 | 5.23 | 0.10 | 6.95 | * |
| gama-cadinene | * | * | 0.26 | * | 4.87 |
| beta-bisabolene | 0.12 | 0.42 | * | * | * |
| gama-murolene | 0.29 | * | * | * | 26.83 |
| isoledene | * | * | * | * | 6.58 |
| gama-gurjunene | * | * | * | * | 5.07 |
| alpha-bergamotene | * | * | * | * | 6.49 |
| beta-seline | * | * | * | * | 2.29 |
| cadina-1(10),4-diene | * | * | * | 1.04 | 7.52 |
| beta-sesquiphellandrene | 8.08 | * | * | * | * |
| germacren B | 0.83 | 1.74 | * | 1.26 | * |
| spathulenol | * | 1.44 | * | * | * |
| aromarderdrene | * | * | 1.05 | * | 2.46 |
| beta-turmerone | 11.71 | * | * | * | * |
| cyperone | * | * | 5.39 | * | * |
| juniper camphor | * | * | * | * | 4.77 |

* not able to detect.

### 3.2. Chemical Determination of Total Polyphenols (as mg/g Dry Matter)

The amount of rutin was significantly highest in samples from Canada goldenrod flowers (20.085 mg/g of sample) and giant goldenrod leaves (10.73 mg/g sample). Leaves from staghorn sumac and tree of heaven were also rich in rutin (more than 4 mg per g of sample). The amount of quercitrin reached 23 g per g of sample from giant goldenrod leaves. Catechyin hydrate reached the highest amount in samples of tree of heaven (21 mg/g sample). All of the detailed amounts are presented in Table 3.

**Table 3.** Average value of polyphenols (mg/g).

| | Canada Goldenrod Leaves | Canada Goldenrod Flowers | Giant Goldenrod Leaves | Giant Goldenrod Flowers | Staghorn Sumac Leaves | Staghorn Sumac Flowers | Bohemian Knotweed Leaves | False Indigo Leaves | Tree of Heaven Leaves | Japanese Knotweed Leaves |
|---|---|---|---|---|---|---|---|---|---|---|
| rutin | 2.822 | 20.085 | 1.031 | 10.727 | 6.514 | 0.200 | 0.291 | 0.631 | 4.552 | 0.097 |
| quercitrin | ND | ND | 23.221 | 3.689 | 4.102 | ND | ND | 0.080 | ND | ND |
| quercetin | 0.081 | 0.545 | 0.150 | 0.471 | 0.006 | 0.035 | 0.241 | 0.049 | 0.002 | 0.107 |
| catechyin hydrate | 3.321 | 7.161 | 15.775 | 5.569 | 4.345 | 11.566 | 4.287 | 4.394 | 21.378 | 4.268 |
| naringin | 0.601 | 8.350 | 1.729 | 7.235 | 24.151 | 5.827 | 1.096 | 0.834 | ND | 0.739 |
| hydroxy coumarin | ND | ND | ND | 0.277 | ND | ND | ND | ND | ND | ND |
| caffeic acid | ND | ND | ND | ND | ND | 0.114 | 0.126 | ND | ND | 0.070 |
| p-coumaric acid | 0.161 | 0.728 | 0.115 | ND | 0.580 | 0.065 | 1.515 | ND | ND | 0.169 |
| ferulic acid | ND | ND | ND | 0.631 | 0.227 | ND | ND | 0.053 | ND | ND |

ND stands for not able to define.

### 3.3. Geochemical Analysis of Wood Ash

According to the analysis, wood ash in our research contained 7.46% $SiO_2$, 2.48% $AlO_3$, 1.74% $Fe_2O_3$, 4.44% $MgO$, 39.55% $CaO$, 0.21% $Na_2O$, 7.54% $K_2O$, 0.16% $TiO_2$, 2.36% $P_2O_5$, 1.26% $MnO$, <0.002% $Cr_2O_3$, 2736 ppm Ba, 50 ppm Ni, 32.10% LOI, 16.2 ppm Co, 1.90 ppm Cs, 3.8 ppm Ga, 668.10 ppm Sr, 1.5 ppm Mo, 167.3 ppm Cu, 29.9 ppm Pb, and 1145 ppm Zn.

### 3.4. Mortality when Plant Powders Were Applied as a Single Use (Single Treatment) (Bioassay 1)

According to Table 4, almost all the main effects and their interactions were significant. There was no significant impact of temperature of the dose of plant powder. Based on general analysis, the mortality of individuals at 20 °C and 25 °C reached 12%. Significantly higher mortality was achieved at 55% R.h., when 18% of all individuals were dead. Regarding the days of exposure, mortality ranged from 8% (day 7) to 15% (day 21). When individuals were exposed to plant powder from false indigo, 15% mortality was detected when exposed to 55% R.h. Additionally, only 4% of weevils were dead at the higher temperature. Higher mortality when exposed to plant powders was, in general, higher with treatments and exposure to the lower R.h. All the values are presented in Figure 1.

**Table 4.** Repeated measures ANOVA (analysis of variance) parameters for the main effects and associated interactions for the mortality level of rice weevil adults (error df = 243).

| Source | df | F | *p* |
|---|---|---|---|
| Source between variables | | | |
| All between | 131 | 340.33 | <0.01 |
| Intercept | 1 | 160.11 | <0.01 |
| Exposure interval | 2 | 115.05 | <0.01 |
| Temperature | 1 | 15.79 | 0.0538 |
| Dose | 1 | 199.27 | 0.0629 |
| Treatment | 10 | 915.26 | <0.01 |
| Exposure interval × temperature | 2 | 5.24 | 0.0054 |
| Exposure interval × dose | 2 | 12.96 | <0.01 |
| Exposure interval × treatment | 20 | 8.75 | <0.01 |
| Temperature × dose | 1 | 41.72 | 0.0529 |
| Temperature × treatment | 10 | 12.71 | <0.01 |
| Dose × treatment | 10 | 14.17 | <0.01 |
| Exposure interval × temperature × dose | 2 | 1.28 | 0.2791 |
| Exposure interval × temperature × treatment | 20 | 1.07 | 0.3706 |
| Exposure interval × dose × treatment | 20 | 0.96 | 0.5124 |
| Temperature × dose × treatment | 10 | 7.15 | <0.01 |
| Exposure interval × temperature × dose × treatment | 20 | 0.86 | 0.6354 |
| Source within variables | | | |
| Within interaction | 112 | 99.88 | <0.01 |
| R.h. | 1 | 776.60 | <0.01 |
| R.h. × exposure interval | 2 | 3.67 | 0.0258 |
| R.h. × temperature | 1 | 129.61 | <0.01 |
| R.h. × dose | 1 | 37.07 | <0.01 |
| R.h. × treatment | 10 | 80.52 | <0.01 |
| R.h. × exposure interval × temperature | 2 | 3.14 | 0.0437 |
| R.h. × exposure interval × dose | 2 | 4.83 | 0.0081 |
| R.h. × exposure interval × treatment | 20 | 11.17 | <0.01 |
| R.h. × temperature × dose | 1 | 128.80 | <0.01 |
| R.h. × temperature × treatment | 10 | 12.78 | <0.01 |
| R.h. × dose × treatment | 10 | 16.38 | <0.01 |
| Exposure interval × R.h. × temperature × dose | 2 | 0.58 | 0.5595 |
| Exposure interval × R.h. × temperature × treatment | 20 | 2.29 | <0.01 |
| Exposure interval × R.h. × dose × treatment | 20 | 0.52 | 0.9593 |
| R.h. × temperature × dose × treatment | 10 | 19.34 | <0.01 |

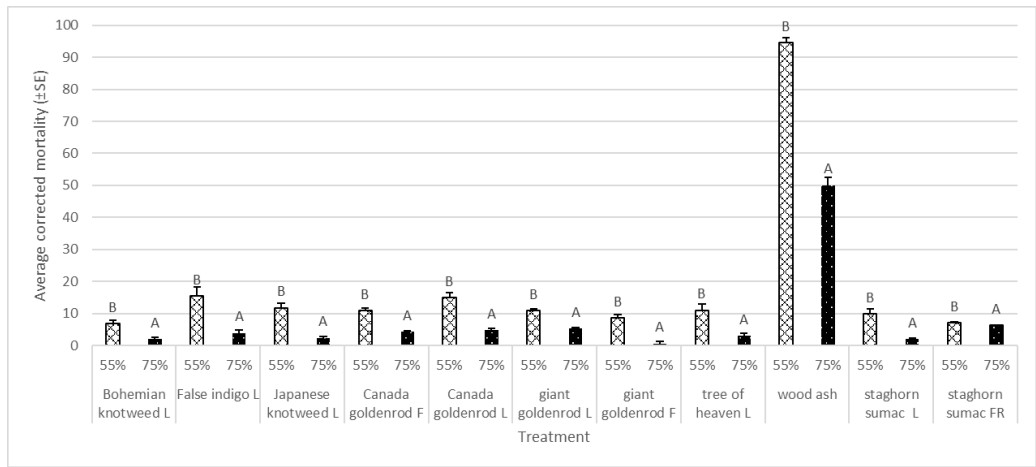

**Figure 1.** Average corrected mortality for different treatments (uppercase letters present differences within treatments between the R.h. parameter).

### 3.5. Mortality when Plant Powders Were Applied for Combined Use (Bioassay 2)

All the main effects and some interactions were significant, according to Table 5. If we compare mortality between temperatures, a significantly higher mortality, 24%, was achieved at 25 °C. A significantly higher mortality, 30%, was also detected at the lower level of R.h. The day 7 post application mortality of individuals reached 7%, day 14 mortality reached 20% and day mortality reached 30%.

**Table 5.** Repeated measures ANOVA of the main effects and associated interactions for the mortality of rice weevil adults in bioassay 2 (df = 168).

| Source | df | F | $p$ |
|---|---|---|---|
| Source between variables | | | |
| All between | 84 | 587.33 | <0.01 |
| Intercept | 1 | 319.57 | <0.01 |
| Exposure interval | 2 | 827.92 | <0.01 |
| Temperature | 1 | 435.68 | <0.01 |
| Treatment | 13 | 364.68 | <0.01 |
| Exposure interval × treatment | 26 | 44.11 | <0.01 |
| Exposure interval × temperature | 2 | 4.14 | <0.01 |
| Temperature × treatment | 13 | 23.64 | <0.01 |
| Exposure interval × temperature × treatment | 26 | 12.64 | <0.01 |
| Source within variables | | | |
| Within interaction | 84 | 785.44 | <0.01 |
| R.h. | 1 | 3008.01 | <0.01 |
| Exposure interval × R.h. | 2 | 323.71 | <0.01 |
| R.h. × temperature | 1 | 233.89 | <0.01 |
| R.h. × treatment | 13 | 270.80 | <0.01 |
| Exposure interval × R.h. × temperature | 2 | 16.71 | <0.01 |
| Exposure interval × R.h. × treatment | 26 | 34.37 | <0.01 |
| R.h. × temperature × treatment | 13 | 14.52 | <0.01 |
| Exposure interval × R.h. × temperature × treatment | 26 | 8.26 | <0.01 |

Day 7 post application, significantly higher mortality was detected in treatments in which the plant powders were enhanced with wood ash. For example, when individuals were exposed to false indigo as a single application at 25 °C and 55% R.h., less than 4% mortality was recorded. When plant powder of false indigo under the same conditions was enhanced with wood ash, 47% mortality was recorded. The highest mortality was recorded when individuals were exposed SilicoSec® (control DE), and less than 47% of individuals were dead. All other values are presented in Table 6.

**Table 6.** Mean mortality (% ± SE) of *Sitophilus oryzae* exposed to different treatments for 7 days.

| | 55% | | 75% | |
|---|---|---|---|---|
| | **20 °C** | **25 °C** | **20 °C** | **25 °C** |
| false indigo | 1.11 ± 0.79 Abc | 3.74 ± 1.18 Bbc | 1.11 ± 0.79 Ab | 2.25 ± 0.97 Abc |
| false indigo–DE | 2.25 ± 0.96 Bc | 2.25 ± 0.79 Bb | 0.74 ± 0.49 Ab | 0.75 ± 0.50 Aa |
| false indigo–wood ash | 7.84 ± 1.75 Be | 47.05 ± 4.24 Bg | 4.44 ± 1.58 Ade | 2.63 ± 1.23 Abc |
| Canada goldenrod | 1.11 ± 0.78 Bbc | 3.76 ± 1.19 Bbc | 1.11 ± 0.56 Ab | 1.91 ± 1.14 Ab |
| Canada goldenrod–DE | 0.00 ± 0.00 Aa | 7.90 ± 2.57 Bd | 2.63 ± 1.47 Bcd | 1.52 ± 1.17 Aab |
| Canada goldenrod–wood ash | 9.75 ± 1.63 Bef | 40.82 ± 4.65 Bf | 2.59 ± 1.08 Acd | 0.75 ± 0.50 Aa |
| control DE | 2.22 ± 1.47 Ac | 46.79 ± 4.71 Bfg | 4.09 ± 1.55 Bd | 5.74 ± 1.99 Ad |
| control wood ash | 7.48 ± 1.43 Be | 38.4 ± 3.44 Bef | 5.22 ± 2.01 Ae | 3.45 ± 1.91 Ab |
| staghorn sumac | 3.03 ± 1.94 Ad | 2.23 ± 1.25 Ab | 2.60 ± 1.21 Acd | 5.85 ± 0.52 Bcd |
| staghorn sumac–DE | 0.00 ± 0.00 Aa | 0.75 ± 0.50 Aa | 2.27 ± 1.15 Bc | 3.56 ± 0.03 Bb |
| staghorn sumac–wood ash | 11.59 ± 2.68 Bf | 38.17 ± 5.30 Bef | 1.87 ± 0.82 Abc | 4.16 ± 1.22 Ac |
| tree of heaven | 1.51 ± 0.60 Bbc | 6.41 ± 1.74 Bc | 0.00 ± 0.00 Aa | 0.74 ± 0.50 Aa |
| tree of heaven–DE | 0.75 ± 0.49 Ab | 3.72 ± 1.41 Bc | 2.96 ± 1.03 Bcd | 1.89 ± 1.29 Aab |
| tree of heaven–wood ash | 12.34 ± 2.47 Ag | 31.30 ± 2.57 Be | 9.37 ± 3.34 Af | 3.89 ± 1.47 Ab |

Within each row and temperature, the mean followed by the same uppercase letter does not differ significantly by the Tukey HSD at $p = 0.05$. For comparisons within rows, df = 1.17 (20 °C, for false indigo, F = 0.00, $p = 1.000$; for false indigo–DE, F = 1.09, $p = 0.3145$; for false indigo–wood ash, F = 2.09, $p = 0.1678$; for Canada goldenrod, F = 3.17, $p = 0.0678$; for Canada goldenrod–DE, F = 3.20, $p = 0.0327$; for Canada goldenrod–wood ash, F = 13.36, $p < 0.01$; for control DE, F = 0.76, $p = 0.0786$; for control wood ash, F = 0.85, $p = 0.3707$; for staghorn sumac, F = 1.39, $p = 0.067$; for staghorn sumac–DE, F = 14.19, $p < 0.01$; for staghorn sumac–wood ash, F = 12.01, $p < 0.01$; for tree of heaven, F = 6.40, $p < 0.01$; for tree of heaven–DE, F = 3.72, $p = 0.0716$; for tree of heaven–wood ash, F = 0.51, $p = 0.4857$; #for 25 °C, false indigo, F = 0.96, $p = 0.3415$, for false indigo–DE, F = 2.56, $p = 0.1294$, for false indigo–wood ash, F = 101.19, $p < 0.01$; for Canada goldenrod, F = 1.28, $p = 0.2739$; for Canada goldenrod–DE, F = 5.11, $p < 0.01$; for Canada goldenrod–wood ash, F = 73.53, $p < 0.01$; for control DE, F = 64.45, $p < 0.01$; for control wood ash, F = 78.90, $p < 0.01$; for staghorn sumac, F = 1.27, $p < 0.01$; for staghorn sumac–DE, F = 66.33, $p < 0.01$; for staghorn sumac–wood ash, F = 77.55, $p < 0.01$; for tree of heaven, F = 20.20, $p < 0.01$; for tree of heaven–DE, F = 5.10, $p = 0.07$; for tree of heaven–wood ash, F = 20.30, $p < 0.01$). Within each column, the mean followed by the same lowercase letter does not differ significantly by the Tukey HSD test at $p = 0.05$ (for comparisons within columns, df = 13.125 (for 20 °C and 55, F = 9.35, $p < 0.01$; for 20 °C and 75% R.h., F = 2.67, $p < 0.01$; for 25 °C and 55% R.h., F = 41.29, $p < 0.01$; for 25 °C and 75% R.h., F = 1.31, $p = 0.0678$)).

Fourteen days post application, less than 15% mortality of individuals was recorded when they were exposed to four different plant powders as a single application. When plant powders of Canada goldenrod and staghorn sumac were enhanced with wood ash, more than 90% mortality was recorded at 25 °C and 55% R.h. All the values are presented in Table 7.

Within each row and temperature, the mean followed by the same uppercase letter does not differ significantly by the Tukey HSD at $p = 0.05$. For comparisons within rows, df = 1.17 (20 °C, for false indigo, F = 0.16, $p = 0.6934$; for false indigo–DE, F = 0.05, $p = 0.8260$, for false indigo–wood ash, F = 69.15, $p < 0.01$; for Canada goldenrod, F = 44.20, $p < 0.01$; for Canada goldenrod–DE, F = 33.82, $p < 0.01$; for Canada goldenrod–wood ash, F = 35.80, $p < 0.01$; for control DE, F = 40.40, $p < 0.01$; for control wood ash, F = 20.30, $p < 0.01$; for staghorn sumac, F = 35.35, $p < 0.01$; for staghorn sumac–DE, F = 15.20, $p < 0.05$; for staghorn sumac–wood ash, F = 45.99, $p < 0.01$, for tree of heaven, F = 14.14, $p = 0.0583$; for tree of heaven–DE, F = 19.18, $p = 0.0698$; for tree of heaven–wood ash, F = 30.20, $p < 0.01$ ## for 25 °C, for false indigo, F = 15.30, $p = 0.0678$; for false indigo–DE, F = 40.80, $p < 0.05$; for false indigo–wood ash, F = 87.33, $p < 0.01$; for Canada goldenrod, F = 5.66, $p = 0.07$; for Canada goldenrod–DE, F = 45.88, $p < 0.01$; for control DE, F = 69.73, $p < 0.01$; for control wood ash, F = 77.66, $p < 0.01$; for staghorn sumac, F = 6.25, $p = 0.0745$; for staghorn sumac–DE, F = 5.55, $p = 0.0765$; for staghorn sumac–wood

ash, F = 10.41, $p < 0.01$)Within each column, the mean followed by the same lowercase letter does not differ significantly by the Tukey HSD test at $p = 0.05$ (for comparisons within columns, df = 13.125 (for 20 °C and 55% R.h. F = 90.98, $p < 0.01$; for 20 °C and 75% R.h., F = 2.98, $p < 0.01$ # for 25 °C and 55% R.h. F = 100.98, $p < 0.01$; for 25 °C and 75% R.h., F = 82.98, $p < 0.01$)).

**Table 7.** Mean mortality (% ± SE) of *Sitophilus oryzae* exposed to different treatments for 14 days.

| | 55% | | 75% | |
| --- | --- | --- | --- | --- |
| | **20 °C** | **25 °C** | **20 °C** | **25 °C** |
| false indigo | 1.51 ± 0.60 Bb | 5.00 ± 2.01 Ac | 1.11 ± 0.78 Aa | 5.48 ± 1.74 Ac |
| false indigo–DE | 3.38 ± 0.97 Ac | 10.52 ± 2.43 Bd | 3.00 ± 1.41 Ab | 2.79 ± 0.84 Aab |
| false indigo–wood ash | 48.97 ± 2.81 Bef | 97.46 ± 1.22 Bi | 10.39 ± 2.39 Ade | 5.87 ± 1.60 Acd |
| Canada goldenrod | 2.00 ± 0.74 Abc | 4.85 ± 1.73 Abc | 4.20 ± 1.88 Bc | 3.41 ± 1.94 Ab |
| Canada goldenrod–DE | 0.00 ± 0.00 Aa | 15.97 ± 2.72 Be | 4.90 ± 1.99 Bc | 2.68 ± 1.23 Aab |
| Canada goldenrod–wood ash | 48.85 ± 3.27 Bef | 96.66 ± 1.00 Be | 3.75 ± 1.64 Abc | 2.27 ± 1.26 Aab |
| control DE | 70.77 ± 3.04 Bh | 100.00 ± 0.00 Bj | 7.54 ± 2.41 Ad | 12.68 ± 5.22 Aef |
| control wood ash | 59.07 ± 6.21 Bg | 86.55 ± 2.51 Bg | 11.49 ± 2.00 Ae | 17.36 ± 4.27 Af |
| staghorn sumac | 3.04 ± 1.94 Ac | 1.88 ± 0.49 Aa | 7.16 ± 1.73 Bcd | 5.30 ± 2.21 Acd |
| staghorn sumac–DE | 5.70 ± 1.37 Bd | 2.30 ± 0.98 Ab | 3.85 ± 1.91 Abc | 2.67 ± 1.23 Ab |
| staghorn sumac–wood ash | 44.36 ± 3.28 Be | 94.24 ± 1.76 Bh | 3.74 ± 1.52 Abc | 11.39 ± 2.83 Ae |
| tree of heaven | 1.90 ± 0.51 Ab | 10.94 ± 3.51 Bd | 3.71 ± 1.32 Bbc | 1.89 ± 1.13 Aa |
| tree of heaven–DE | 3.83 ± 1.67 Ac | 8.25 ± 0.98 Bcd | 4.49 ± 2.19 Bc | 4.31 ± 2.71 Ac |
| tree of heaven–wood ash | 51.63 ± 2.93 Bef | 69.23 ± 8.84 Bf | 13.43 ± 3.65 Af | 5.92 ± 1.63 Ad |

After day 21, 100% mortality was recorded when rice weevil adults were exposed to wood ash and SilicoSec® at 25 °C and 55%. In general, higher mortality levels were detected at the lower R.h. value. All the data are presented in Table 8. More than 80% of individuals were dead when exposed to the lower R.h. and the four different plant powders enhanced with wood ash.

**Table 8.** Mean mortality (% ± SE) of *Sitophilus oryzae* exposed to different treatments for 21 days.

| | 55% | | 75% | |
| --- | --- | --- | --- | --- |
| | **20 °C** | **25 °C** | **20 °C** | **25 °C** |
| false indigo | 3.04 ± 1.46 Aab | 9.49 ± 10.07 Bb | 3.07 ± 1.52 Aab | 5.64 ± 2.19 Ac |
| false indigo–DE | 27.40 ± 3.58 Bc | 52.09 ± 5.53 Ae | 3.04 ± 1.43 Aab | 3.97 ± 2.63 Ab |
| false indigo–wood ash | 84.57 ± 3.50 Be | 99.54 ± 0.46 Bgh | 6.38 ± 2.23 Ab | 15.16 ± 3.43 Ae |
| Canada goldenrod | 2.27 ± 1.25 Aab | 6.55 ± 2.19 Aa | 4.27 ± 2.03 Bab | 5.00 ± 2.24 Abc |
| Canada goldenrod–DE | 4.62 ± 1.85 Ab | 37.12 ± 10.67 Ad | 5.40 ± 2.03 Bab | 90.07 ± 4.50 Bg |
| Canada goldenrod–wood ash | 82.37 ± 3.15 Bde | 99.12 ± 0.54 Bg | 5.19 ± 3.41 Aab | 11.18 ± 3.97 Ad |
| control DE | 97.33 ± 1.25 Bg | 100.00 ± 0.00 Bh | 10.57 ± 3.89 Ac | 12.49 ± 3.48 Ade |
| control wood ash | 78.97 ± 4.85 Bd | 100.00 ± 0.00 Bh | 29.75 ± 3.78 Ae | 41.02 ± 4.99 Af |
| staghorn sumac | 4.03 ± 2.09 Aab | 6.10 ± 2.50 Ba | 2.31 ± 1.42 Aa | 8.29 ± 0.59 Aa |
| staghorn sumac–DE | 5.70 ± 1.02 Ab | 18.49 ± 4.81 Bc | 7.29 ± 2.27 Bbc | 9.97 ± 1.33 Aa |
| staghorn sumac–wood ash | 81.21 ± 4.47 Bde | 100.00 ± 0.00 Bh | 4.57 ± 2.14 Aab | 13.34 ± 4.55 Ae |
| tree of heaven | 1.90 ± 0.82 Aa | 8.46 ± 2.64 Bab | 5.54 ± 2.55 Bab | 1.56 ± 1.17 Aab |
| tree of heaven–DE | 5.20 ± 1.72 Ab | 33.17 ± 5.67 Bd | 5.57 ± 2.83 Bab | 4.57 ± 2.66 Abc |
| tree of heaven–wood ash | 90.05 ± 2.08 Bf | 90.01 ± 5.75 Bf | 15.06 ± 4.23 Ad | 8.30 ± 2.90 Acd |

Within each row and temperature, the mean followed by the same uppercase letter does not differ significantly by the Tukey HSD at $p = 0.05$. For comparisons within rows, df = 1.17 (20 °C, for false indigo, F = 88.10, $p < 0.01$; for false indigo–DE, F = 140.18, $p < 0.01$; for false indigo-wood ash, F = 99.45, $p < 0.01$; for Canada goldenrod, F = 120.13, $p < 0.01$; for Canada goldenrod–DE, F = 99.14, $p < 0.01$; for Canada goldenrod–wood ash, F = 200.18, $p < 0.01$; for control DE, F = 155.15, $p < 0.01$; for control wood ash, F = 97.14, $p < 0.01$; for staghorn sumac, F = 180.30, $p < 0.01$; for staghorn sumac–DE, F = 130.11, $p < 0.01$; for staghorn sumac–wood ash, F = 111.10, $p < 0.01$; for three of heaven, F = 21.13, $p < 0.01$; for three of heaven–DE, F = 150.22, $p < 0.01$; for tree of heaven–wood ash,

F = 145.22, $p$ < 0.01 ## 25 °C, for false indigo, F = 71.40, $p$ < 0.01; for false indigo–DE, F = 79.18, $p$ < 0.01; for false indigo-wood ash, F = 88.88, $p$ < 0.01; for Canada goldenrod, F = 101.93, $p$ < 0.01; for Canada goldenrod–DE, F = 112.14, $p$ < 0.01; for Canada goldenrod–wood ash, F = 163.18, $p$ < 0.01; for control DE, F = 120.15, $p$ < 0.01; for control wood ash, F = 130.14, $p$ < 0.01; for staghorn sumac, F = 80.40, $p$ < 0.01; for staghorn sumac–DE, F = 120.11, $p$ < 0.01; for staghorn sumac–wood ash, F = 121.10, $p$ < 0.01; for three of heaven, F = 166.13, $p$ < 0.01; for three of heaven–DE, F = 150.22, $p$ < 0.01; for tree of heaven–wood ash, F = 145.22, $p$ < 0.01). Within each column, the mean followed by the same lowercase letter does not differ significantly by the Tukey HSD test at $p$ = 0.05 (for comparisons within columns, df = 13.125 (for 20 °C and 55% R.h. F = 120.17, $p$ < 0.01; for 20 °C and 75% R.h., F = 130.78, $p$ < 0.01 # for 25 °C and 55% R.h. F = 61.13, $p$ < 0.01; for 25 °C and 75% R.h., F = 82.98, $p$ < 0.01)).

### 3.6. Progeny Production for Bioassay 2

According to the analysis, the progeny production of rice weevil was affected by temperature (F = 320.17; $p$ < 0.01), R.h. level (F = 278.10, $p$ < 0.01) and treatment (F = 303.11, $p$ < 0.01). All the values are presented in Table 9. Progeny production was significantly highest in the untreated control.

**Table 9.** Progeny production of Sitophilus oryzae.

| | 55% | | 75% | |
|---|---|---|---|---|
| | **20 °C** | **25 °C** | **20 °C** | **25 °C** |
| false indigo | 140.55 ± 8.33 Ah | 65.65 ±11.10 Agh | 140.88 ± 1.44 Af | 145.17 ± 15.77 Bj |
| false indigo–DE | 68.33 ± 5.12 Ad | 25.11 ±3.50 Ae | 130.88 ± 1.36 Be | 139.12 ± 16.88 Bi |
| false indigo–wood ash | 23.12 ± 7.65 Abc | 11.88 ±2.77 Ac | 115.88 ± 9.12 Bd | 100.11 ± 12.66 Bf |
| Canada goldenrod | 136.36 ± 5.88 Bh | 70.66 ±5.55 Af | 108.55 ± 12.14 Acd | 130.77 ± 15.66 Bh |
| Canada goldenrod–DE | 115.91 ± 7.33 Afg | 28.12 ±3.55 Ae | 100.66 ± 14.12 Acd | 25.88 ± 2.88 Aa |
| Canada goldenrod–wood ash | 17.66 ± 2.33 Aab | 8.13 ±1.32 Ab | 99.17 ± 5.99 Ac | 100.88 ± 8.88 Bf |
| control DE | 11.30 ± 2.74 Aa | 1.02 ±0.04 Aa | 80.16 ± 6.66 Bb | 90.15 ± 5.66 Be |
| control wood ash | 21.13 ± 2.97 Ab | 1.05 ±0.06 Aa | 40.12 ± 5.66 Ba | 30.77 ± 5.66 Bb |
| staghorn sumac | 120.11 ± 7.99 Ag | 95.11 ±6.66 Bi | 130.88 ± 5.56 Ae | 65.11 ± 5.12 Ad |
| staghorn sumac–DE | 80.40 ± 2.55 Ae | 45.44 ±5.44 Af | 115.15 ± 6.65 Bd | 68.56 ± 5.10 Bd |
| staghorn sumac–wood ash | 30.30 ± 1.66 Ac | 2.15 ±1.22 Aa | 130.77 ± 7.69 Be | 50.42 ± 5.12 Bc |
| tree of heaven | 136.44 ± 12.12 Ah | 79.88 ±2.23 Ah | 114.22 ± 6.77 Acd | 160.88 ± 5.99 Bk |
| tree of heaven–DE | 100.21 ± 6.55 Af | 57.33 ±1.77 Ag | 144.15 ± 5.66 Bf | 130.87 ± 6.12 Bh |
| tree of heaven–wood ash | 14.25 ± 1.99 Aa | 15.24 ±1.22 Ad | 80.13 ± 6.66 Bb | 120.88 ± 5.67 Bg |
| control untreated grain | 170.22 ± 4.33 Ai | 190.66 ±12.11 Aj | 380.11 ± 25.23 Bg | 430.56 ± 12.81 Be |

Within each row and temperature, the mean followed by the same uppercase letter does not differ significantly by the Tukey HSD at $p$ = 0.05. For comparisons within rows, df = 1.17 (20 °C, for false indigo, F = 211.10, $p$ = 0.06; for false indigo–DE, F = 190.38, $p$ < 0.01; for false indigo-wood ash, F = 199.15, $p$ < 0.01; for Canada goldenrod, F = 180.13, $p$ < 0.01; for Canada goldenrod–DE, F = 103.14, $p$ = 0.06; for Canada goldenrod–wood ash, F = 150.18, $p$ < 0.01; for control DE, F = 135.15, $p$ < 0.01; for control wood ash, F = 97.14, $p$ < 0.01; for staghorn sumac, F = 160.30, $p$ < 0.01; for staghorn sumac–DE, F = 130.11, $p$ = 0.04; for staghorn sumac–wood ash, F = 111.10, $p$ < 0.01; for three of heaven, F = 21.13, $p$ = 0.06; for three of heaven–DE, F = 150.22, $p$ < 0.01; for tree of heaven–wood ash, F = 203.22, $p$ < 0.01 ## 25 °C, for false indigo, F = 104.40, $p$ < 0.01; for false indigo–DE, F = 100.18, $p$ < 0.01; for false indigo-wood ash, F = 99.88, $p$ < 0.01; for Canada goldenrod, F = 101.93, $p$ < 0.01; for Canada goldenrod–DE, F = 182.14, $p$ = 0.07; for Canada goldenrod–wood ash, F = 168.18, $p$ < 0.01; for control DE, F = 170.15, $p$ < 0.01; for control wood ash, F = 180.14, $p$ < 0.01; for staghorn sumac, F = 180.30, $p$ < 0.01; for staghorn sumac–DE, F = 120.11, $p$ < 0.01; for staghorn sumac–wood ash, F = 121.10, $p$ < 0.01; for three of heaven, F = 166.13, $p$ < 0.01; for three of heaven–DE, F = 160.22, $p$ < 0.01; for tree of heaven–wood ash, F = 165.22, $p$ < 0.01). Within each column, the mean followed by the same lowercase letter does not differ significantly by the Tukey HSD test at $p$ = 0.05 (for comparisons within columns, df = 13.125 (for 20 °C and 55% R.h. F = 203.97, $p$ < 0.01; for 20 °C and 75% R.h., F = 200.78, $p$ < 0.01 # for 25 °C and 55% R.h. F = 161.13, $p$ < 0.01; for 25 °C and 75% R.h., F = 162.98, $p$ < 0.01)).

## 4. Discussion

Our study focused on analysis of the efficacy of plant powders from invasive alien plant species against rice weevil. Plant powders were applied alone or enhanced with two other inert dusts, diatomaceous earth and wood ash. Our study found that the mortality of individuals was significantly influenced by relative humidity and temperature. The same held true for treatments in which we used milled leaves of alien invasive plants, as well as for treatments in which we studied the synergistic effects of milled leaves with the two types of inert dust at a reduced dose. The fact that increased temperature and decreased relative humidity improved the effects (higher mortality of beetles) of inert dusts (wood ash, diatomaceous earth, zeolites ... ) was confirmed by [4,15,16]. It is known, for inert dusts, that they have the highest efficacy at low humidity [17] because they cause desiccation. Insects can lose water because the dusts remove the waxy layer of the cuticule by adsorption. Low R.h. was the main factor in treatments, where we have added wood ash and diatomaceous earth. To date, studies of the effects of milled plant material in independent applications have also reported low mortality of beetles [5], as established in the first part of our research. Due to the already known negative influence of inert dust on stored grains [17], reducing the quantity of inert dust is essential.

Synergistic effects of inert dusts combined with plant insecticides have already been confirmed by some studies, and the effects are primarily connected with the use of essential oils [1], as well as the use of plant powders. Combinations of essential oils and diatomaceous earth are supposed to cause additional stress to storage insect pests, as essential oils increase the movement of pests and consequently facilitate the effects of diatomaceous earth. It is confirmed [18] that adding diatomaceous earth to powders from *Piper guineense* and *Senna siamea* improved the effects of the powders and thus achieved higher mortality than from their independent application. At lower humidity and higher temperature, our study achieved a comparable mortality (over 90%) during treatments in which a lower concentration of ash was added to powders (bioassay 2), and in the treatment in which we used wood ash at a higher concentration. Comparable insecticidal effects during the single application of wood ash and the application of combinations of wood ash and four powders from IAPs were also achieved at the lower temperature and lower humidity, yet the mortality after the 21st day of exposure did not exceed 95%. At the higher relative humidity, we achieved mortality in individual treatments that was higher than 40% (control treatment–wood ash) and 90% (a combination of wood ash and Canada goldenrod) only after 21 days. Adding diatomaceous earth to the milled leaves of IAPs improved the insecticidal properties of the plant powders, though not as significantly as when wood ash was added. The combined use of plant insecticides (milled plant parts) and wood ash was recommended in the study by [19], and this application is supposed to be primarily useful for farms in less developed parts of the world, where this treatment would considerably lower the costs of pest suppression. To date, studies on the efficacy of wood ash for the suppression of storage pests have not confirmed their negative effects on stored wheat [3], and the mortality of beetles has also not been shown to be affected by the concentration of wood ash [4]. On the other hand, an increasing number of studies have proven that it makes sense to reduce the quantity of diatomaceous earth in treatments [1,17] because of its demonstrated negative effects, such as its adverse effects on the physical and mechanical properties of grain. For this reason, our study involved further decreases in the doses of the selected inert dusts, i.e., diatomaceous earth and spruce wood ash (relative to previous doses). The independent application of diatomaceous earth in our study produced high mortality, which was already established by [20]. The combination of diatomaceous earth and milled leaves of alien invasive plants did not produce high mortality of rice weevil. The importance of the duration of exposure to inert dust was shown both in our study and in some previous studies [4,20,21].

The chemical composition of Norway spruce wood ash in our study differed from the chemical composition of Norway spruce wood ash that we used in a previous study [4]. We proved that the chemical composition of plants is also influenced by their location site. In both cases, it holds true that the use of wood ash influenced the progeny production of beetles. The combinations of plant powder and wood ash produced satisfactory insecticidal effects on progeny production. The best insecticidal

effects on progeny production were achieved also by the independent use of wood ash or its use in combination with diatomaceous earth at a lower humidity [4] and higher temperature.

Our research is among the first to also present the chemical compositions of the selected invasive plants and their effects on storage pests. Among plant insecticides used for the suppression of storage pests, studies to date have confirmed the highest efficacy of essential oils [1]. Staghorn sumac, Bohemian knotweed, and tree of heaven contain very little essential oil in their leaves, which consequently decreases the possibility of their use as insecticides. We did not detect essential oils in leaves of Japanese knotweed. Leaves of Canada goldenrod, which were included due to their easy accessibility in both parts of the research, contain the most essential oil per gram of sample. The total polyphenols content is markedly high in blossoms of Canada and giant goldenrod. With independent application of plant powders, Canada goldenrod (leaves and blossoms) produced the highest mortality. With regard to the applied inert dusts, we can point to the contents of $SiO_2$ [22] or pronounced hydrophilicity [4] as factors underlying the insecticidal effects, yet the invasive alien plant species that displayed the highest insecticidal efficacy in mixtures with inert dusts will require more detailed research into its chemical composition, particularly the influence of the age of plants on the contents of essential oils and polyphenols.

Essential oils, especially alfa-pinene, beta-pinene, alpha-phellandrene, ocimene, borneol, germacrene-B, and gama-cadinine, which were found in invasive plant species in our research, are important components of spice plants (Lauraceae) and are often used to control stored product pests in some regions [23]. The essential oils mentioned above are also important components of some other Mediterranean plants, such as *Citrus bergamia*, *Lavandula hybrida*, *Foeniculum vulgare* [24], which are often used against stored product pests. Rutin, which is also present in *Moringa oleifera* and acts as a repellent against *Sitophilus zeamais* [25], was present in samples in our research. However, the content of the mentioned substances in invasive alien plants in our research was too low to achieve higher insecticidal efficacy.

## 5. Conclusions

While studying the synergistic effects of different types of powders with the aim to overcome the disadvantages of using inert dusts (diatomaceous earth and wood ash) as a single use treatment, we established that mixing invasive alien plant powders with both inert dusts does not improve their efficacy in controlling rice weevil. Obviously, the contents of essential oils and polyphenols as the most important constituents of the insecticidal action of invasive alien plants were too low for single or mixed use of their powders. However, we expect that a higher insecticidal efficacy of invasive alien plants could be achieved by preparing essential oils from the studied plant parts, and we will investigate this in our further research.

**Author Contributions:** T.B.—project administration, data curation, and formal analysis, writing—original draft, writing—review and editing; investigation; A.H.—formal analysis; M.O.—methodology; I.J.K.—methodology; K.R.—methodology; S.T.—conceptualization, resources, supervision, validation, visualization, and writing—review and editing. All authors have read and agreed to the published version of the manuscript.

**Funding:** The research was co-funded by European Regional Developmental Fund through the Urban Innovative Actions Initiative, grant number UIA02_228.

**Acknowledgments:** Research was funded by the project ApPLAuSE (Alien PLant SpEcies)—from harmful to useful with citizen-led activities. Applause is a project of UIA (Urban Innovative Actions), an Initiative of the European Union that provides Europe with resources to test new and unproven solutions to address urban challenges. The three-year project is financed by the European Regional Development Fund. The authors are responsible for all the information stated in this paper. UIA is not responsible for their use. Andrija Vasilić and Andraž Stibilj are acknowledged for their technical assistance. We extend our sincere thanks to Susie Woo from Bureau Veritas Commodities Canada Ltd. for providing descriptions of the methods and equipment used for geochemical analysis of wood ash.

**Conflicts of Interest:** The authors declare no conflict of interest. The funders had no role in the design of the study, in the collection, analyses or interpretation of data, in the writing of the manuscript or in the decision to publish the results.

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
