# Peer review of "The First Evidence of the Insecticidal Potential of Plant Powders from Invasive Alien Plants against Rice Weevil under Laboratory Conditions"

_applsci, doi:10.3390/app10217828_

Round 1
Reviewer 1 Report
The evaluated work deals with analysis of the efficacy of invasive alien species against rice weevil. This review is very unique since it is among the first in which the chemical composition of the invasive plants are displayed and it takes into account their effects alone and in the mixture with inert dust against stored pest.
Thus, this research represents a significant contribution to the insight of the inert dust usage.
Although, the discussion is concise and clear, I recommend additional discussion about the mortality rate according to the RH and temperature level. Author should state why the mortality was higher at lower humidity and higher temperature levels, since it was evident in most of the treatments.
Also there are some suggestions:
Keywords: I suggest a smaller number of keywords (up to 10), and I am not sure to which the keyword applause refers?
Superscript letters for date should be changed: line 89, 90 and 91, and 4th in the line 92, should be changed into 4th
Line 182 unit is missing... 500 ?? of winter wheat
Line 205 Label ® for SilicoSec should be written uniformly
Line 399 Our study focused on analysing the...should be change into ...focused on analysis of the...
References some number of volumes should be written in Italic form (line 529, 551, 559, 521, 565)-
Line 581 The publishing year after the list of authors should be deleted
Author Response
The evaluated work deals with analysis of the efficacy of invasive alien species against rice weevil. This review is very unique since it is among the first in which the chemical composition of the invasive plants are displayed and it takes into account their effects alone and in the mixture with inert dust against stored pest.
Thus, this research represents a significant contribution to the insight of the inert dust usage.
Although, the discussion is concise and clear, I recommend additional discussion about the mortality rate according to the RH and temperature level. Author should state why the mortality was higher at lower humidity and higher temperature levels, since it was evident in most of the treatments.
Comment: It is known for inert dusts, that they have the highest efficacy at low humidity [17], because they cause desiccation. Insects can loose water, because the dusts remove the waxy layer of the cuticule by adsorption. Low R.h. was the main key in treatments, where we have added wood ash and diatomaceous earth. Lines: 414-418
Also there are some suggestions:
Keywords: I suggest a smaller number of keywords (up to 10), and I am not sure to which the keyword applause refers?
Comment: Corrected as instructed. Applause refers to the name of the project. Research was funded within the project. Lines 49-51. Applause stands for the name of the project.
Superscript letters for date should be changed: line 89, 90 and 91, and 4th in the line 92, should be changed into 4th
Comment: Corrected as instructed. Lines 87-93.
Line 182 unit is missing... 500 ?? of winter wheat
Comment: Corrected as instructed. Line 188.
Line 205 Label ® for SilicoSec should be written uniformly
Comment: Corrected as instructed. Line 213.
Line 399 Our study focused on analysing the...should be change into ...focused on analysis of the...
Comment: Corrected as instructed. Line 407.
References some number of volumes should be written in Italic form (line 529, 551, 559, 521, 565)-
Comment: Corrected as instructed. Lines 533, 553,565,573,577.
Line 581 The publishing year after the list of authors should be deleted
Comment: Corrected as instructed. Line 587

Reviewer 2 Report
Bhohinc et al. investigated the effect of dry powder of a suite of invasive plant species to test their insecticidal effect on rice beettle. The study may be of interest not only from the scientific community but also from industry. Topic of the manuscript is timely, the applied methods are adequate. However I have some concerns about the ms, which strengthen me to advise a Major Revision.
Introduction is a bit short and concise, but reader-friendly and interesting, which make reading enjoyable.
Methods
L85-93: Why did you collect leaves in different periods? Did you take the status of each species into consideration? Please write it!
L95: „the dried material was milled with a mill” what was the targeted particle size of the powders? Was that equal to each species, to the study media (wheat flour) and positive control (Picea abies ash)? Differences in particle size could also have effect on the presented results.
HPLC and GC analyses: why did you study these exact active ingredients? I suggest it is because they were previously proved to been found and effective of those species. Please explain to the reader.
L165: “Rice weevils were 2-4 weeks old.” Could you please describe what is that mean for the species? Were they larvae, pupa, or adults?
L185: “Thirty individuals were placed into 60-ml flask” Were these flasks closed or open? This could affect the transfer of weevil or the effect of humidity.
210-212: “Mortality counts were analysed by using repeated measures MANOVA with exposure day as the repeated measures variable and treatment, temperature and R.h. as the main effects” MANOVA is actually not a correct option for count data, because you have to face with heteroskedasticity, skewness, and discreteness. You should apply binomial or Poisson GLM (Generalized linear model) analyses, or maybe you can set it up as a chi-square test. This is same for the ANOVA (progeny) testing.
Results
For the Bioassay 1 it would be nice to present pairwise comparisons too.
Author Response
@
Open Review
Introduction is a bit short and concise, but reader-friendly and interesting, which make reading enjoyable.
Methods
L85-93: Why did you collect leaves in different periods? Did you take the status of each species into consideration? Please write it!
Comment: All tested invasive plant species were collected in all given dates. Due to the fact, that several research were done in same time interval, this research contained material from several days. Additional explanations has been added to the text. Lines: 93-95.
L95: „the dried material was milled with a mill” what was the targeted particle size of the powders? Was that equal to each species, to the study media (wheat flour) and positive control (Picea abies ash)? Differences in particle size could also have effect on the presented results.
Comment: We appreciate given comment. The particle size of powders (plant powders, wood ash) were between 20-200 µm (line 100), while particle size of SilicoSec ® was 2-18 µm (line 205).
HPLC and GC analyses: why did you study these exact active ingredients? I suggest it is because they were previously proved to been found and effective of those species. Please explain to the reader.
Comment: We have taken into consideration your comment. Active ingredients that were detected in our survey where chosen according to detailed study of scientific literature and evidence of possible insecticidal efficacy. Lines 103-104
L165: “Rice weevils were 2-4 weeks old.” Could you please describe what is that mean for the species? Were they larvae, pupa, or adults?
Comment: This term »rice weevils« is related to rice weevil adults. It was corrected and additionally explained in the text. Line 171.
L185: “Thirty individuals were placed into 60-ml flask” Were these flasks closed or open? This could affect the transfer of weevil or the effect of humidity.
Comment: Flusks were covered with mesh to prevent rice weevil adults from escaping. It has been additonally explained in the text. Lines 185-186. it is standard methodology, that has been previously used in papers, Bohinc et al. (2018) and Bohinc et al. (2020); and in also in many other papers (Rojht et al., 2010).
210-212: “Mortality counts were analysed by using repeated measures MANOVA with exposure day as the repeated measures variable and treatment, temperature and R.h. as the main effects” MANOVA is actually not a correct option for count data, because you have to face with heteroskedasticity, skewness, and discreteness. You should apply binomial or Poisson GLM (Generalized linear model) analyses, or maybe you can set it up as a chi-square test. This is same for the ANOVA (progeny) testing.
Comment: we appreciate your comment Statistical analysis were done according to reviewers suggestions for journal »Journal of Stored Product Research«, where we have published two of our previous work. Identical statistical analysis was done in some other papers, that are related to the topic that is presented in our paper.
Bohinc, T.; Horvat, A.; Andrić, G.; Pražić Golić, M.; Kljajić, P.; Trdan, S. Comparison of three different wood ashes and diatomaceous earth in controlling the maize weevil under laboratory conditions. J. Stored Prod. Res. 2018, 79, 1-8.
Bohinc, T.; Horvat, A; Andrić, G.; Prazić Golić, M.; Kljajić, P.; Trdan, S. Natural versus synthetic zeolites for controlling the maize weevil (Sitophilus zeamais) – like Messi versus Ronaldo? J. Stored Prod. Res. 2020, https://doi.org./10.1016/j.jspr.2020.101639.
Eroglu, N.; Sakka, M.K.; Emekci, M.; Athanassiou, C.G. Effect of zeolite formulations on the mortality and progeny production of Sitophilus oryzae and Oryzaephilus surinamensis at different temperature and relative humidity levels. J. Stored Prod. Res. 2019, 81, 40-45.
Georgousakis, C., Sakka, M.K., Karkanis, A.C., Athanassiou, C.G. Gone with the weed: Population growth of Sitophilus oryzae and Rhyzopertha dominica in wheat and barley containing seeds of Silybum marianum. J Stored Prod. Res. 2020. http://doi.org./10.1016/j.jspr.2020.101602
Results
For the Bioassay 1 it would be nice to present pairwise comparisons too.
Comment: We appreciate your comment. According to detailed discussion with all co-authors, we have decided, that we will not expose very low (below 10 %) corrected mortality in Bioassay 1. For these cause, we have decided to focus on biassay 2.

Round 2
Reviewer 2 Report
Bohinc et al. revised the manuscript according to my previous suggestions which made the manuscript much clearer to understand. I welcome these changes and now suggest its aception to Applied Sciences.